# Difficulties in Diagnosing Extraperitoneal Ureteroinguinal Hernias: A Review of the Literature and Clinical Experience of a Rare Encounter in Acute Surgical Care Settings

**DOI:** 10.3390/diagnostics12020353

**Published:** 2022-01-29

**Authors:** Catalin Pirvu, Stelian Pantea, Alin Popescu, Mirela Loredana Grigoras, Felix Bratosin, Andrei Valceanu, Tudorel Mihoc, Vlad Dema, Mircea Selaru

**Affiliations:** 1Department of General Surgery, “Victor Babes” University of Medicine and Pharmacy, 300041 Timisoara, Romania; pirvu.catalin.alexandru@gmail.com (C.P.); pantea.stelian@umft.ro (S.P.); felix.bratosin7@gmail.com (F.B.); apvalceanu@yahoo.com (A.V.); tudorel_mihoc@yahoo.com (T.M.); selaru_mircea@yahoo.com (M.S.); 2Department of Obstetrics and Gynecology, “Victor Babes” University of Medicine and Pharmacy, 300041 Timisoara, Romania; alinp22@yahoo.com; 3Department of Anatomy and Embryology, “Victor Babes” University of Medicine and Pharmacy, 300041 Timisoara, Romania; 4Department of Urology, “Victor Babes” University of Medicine and Pharmacy, 300041 Timisoara, Romania; vlad.dema@yahoo.com

**Keywords:** hernia diagnosis, CT urography, ureteral obstruction, ureteroinguinal hernia, inguinal surgery

## Abstract

Although inguinal hernia repair is one of the most common surgical procedures, finding a retroperitoneal structure, such as the ureter, is a rather rare occurrence. Ureteroinguinal hernias may arise in the presence or absence of obstructive uropathy, the latter raising difficulties in diagnosis for the general surgeon performing a regular inguinal hernia surgery. This study aims to collect the relevant literature describing the diagnosis and management of ureteroinguinal hernias and update it with a case encountered in our clinic. The following study was reported following the SCARE guidelines. The relevant literature describes less than 150 cases of ureteroinguinal hernias overall, considering the 1.7% prevalence of inguinal hernias in the general population. With only 20% of these hernias being described as extraperitoneal, such an encounter becomes an extremely rare finding. Our clinical experience brings a case of a 75-year-old male with frequent urinary tract infections and a large irreducible inguinoscrotal hernia of about 20/12 cm located at the right scrotum. The patient underwent an open inguinal hernia repair technique under general anesthesia, incidentally finding an extraperitoneal ureteral herniation. Segmental ureterectomy was performed with uneventful recovery. Intraoperatively, finding an incidental ureteroinguinal hernia raises concerns about probable urinary tract complications during regular hernia repair surgery and whether the diagnosis is likely to happen prior to surgical intervention. Although imaging is rarely indicated in inguinal hernias, the case reports show that a pelvic CT scan with urography in symptomatic patients with urinary symptoms will provide accurate confirmation of the diagnosis. The relevant literature is limited due to the rarity of respective cases, thus making standardized management of such cases unlikely.

## 1. Introduction

Inguinal hernias are classified as direct and indirect based on the relationship of the herniation sac with the inguinal canal [1], while both types can involve structures from the intraperitoneal or extraperitoneal space [2]. A direct hernia is formed when the hernia sac protrudes directly through the posterior wall of the inguinal canal [3,4], while in the indirect type, the sac follows the inguinal canal in conjunction with the spermatic cord after entering through the internal inguinal ring [5,6]. The prevalence of inguinal hernias accounts for 1.7% of the general population, with an increasing 4% in people older than 45 [7]. Relative to the whole range of abdominal wall hernias, the inguinal type comprises around 75% of the reported cases, where women have a 3% lifetime risk [8] and a much higher risk of 27% is present in men [9]. Studies report around 95% of patients presenting to primary care with inguinal hernias to be men, with a high incidence of over 75 year olds [10] at around 20 per thousand person–years [11]. Even though surgeons encounter many cases of inguinal hernias that are easily diagnosed based on the clinical picture and clinical experience, the rate of misdiagnosed or undiagnosed inguinal hernias can go up to 8%, as some studies report [12,13]. Overall, the surgery for repairing inguinal hernias is considered one of the most common procedures in the surgical field [14,15,16], where available data shows 10 surgeries performed per 100,000 people in the U.K., and up to a higher number of 28 surgeries performed per 100,000 people in the U.S. [7].

Inguinal hernias containing a ureteral segment are an extremely rare finding in patients with native kidneys when there is no ectopic or transplanted kidney involved. To date, fewer than 140 cases of ureteral inguinal hernias have been described [17], and around 8 cases of ureteroinguinal hernias were described in detail by the existing literature in patients with native kidneys [18]. By etiology, ureteroinguinal hernias can be spontaneous, postoperative, or kidney transplantation complications [19]. Regardless of prevalence, ureteric hernias are classified by the anatomical region involved as inguinal, femoral, thoracic, and para-iliac [20,21], where studies report around 40% of them in the inguinal region [22]. The inguinal type has 2 anatomical variants: paraperitoneal, counting 80% of cases, and extraperitoneal, with just 20% [23]. The most prevalent type is similar to sliding hernias, being formed when the peritoneal sac is adherent to the ureter, dragging it while descending into forming the herniation process. The paraperitoneal type is most commonly an indirect hernia, direct hernias being a very rare occurrence [24], where 94% of the patients are men showing a 67% proportion of presentations on the right side [19]. The herniation of the ureter on the left side rarely occurs due to the position of the sigmoid colon on the left side [25]. More frequently, it occurs in the sixth decade of life [26], although there have been cases reported in patients as young as six weeks old [27,28]. There is no herniation sac in the least common type since the ureter descends along with retroperitoneal fat [29].

The first documented case of ureteroinguinal hernia was in 1880 during an autopsy [30], while, ever since that incident, the literature remains scarce in documented cases and their diagnosis. The extraperitoneal type of ureteroinguinal hernia is believed to occur due to developmental anomalies of the ureter from the Wolffian duct. When the testes descend, they apply traction on the ureter through the persistence of two genitoinguinal ligaments [31]. The Wolffian duct connects these ligaments to the testicles. In the third or fourth week of gestation, the Wolffian duct emits a pedicle that becomes a ureter [32,33]. Therefore, if there is insufficient ureteral differentiation, the ureter may be easily pulled along with the testicle into the scrotum and may herniate [34].

Ureteroinguinal hernias are most frequently found in obese patients or people who have undergone a kidney transplant [35]. In contrast, the general high-risk conditions associated with inguinal hernias with urological findings comprise the male gender [36], an age older than 50, obesity [37,38], urological malignancies [39,40,41], and benign prostatic hyperplasia [42,43,44]. Even though some conditions are frequently found together with a ureteroinguinal hernia, they are not statistically proven to be predisposing factors; thus, a pre-operative diagnosis attempt for a ureteral herniation will often be unjustified or disregarded. Consequently, in the event of such an occurrence, the risk of damaging the ureter is increased, where urinary leakage is seen from the wounded ureter [45,46,47]. Therefore, a ureteroraphy or segmental ureterectomy with anastomosis should be performed [48,49]. Similarly, in such cases where the ureter was identified before damaging it, there can still be no option to avoid the ureterectomy due to the impossibility of manipulating it out of the herniation process. In less complicated cases, definitive treatment can be undertaken and obtained by the surgical reduction of the ureter into the abdominal cavity and herniorrhaphy [50,51].

A tortuous ureter is often described in imaging studies and might suggest a ureteroinguinal hernia associated with the patient’s clinical picture. Tortuous ureters are usually found in ectopic kidneys [52,53], the majority being situated lower, in the pelvis [54], with an incidence up to 1 in 3000 cases, caused by an incomplete ascent of the kidney during the embryonic period. Thus, ultrasound findings of renal ectopia with tortuous ureters are predisposed to ureteral herniation [55]. In transplanted kidneys, the same event can occur where a malpositioned new kidney placed in the lower abdomen [56] can allow its ureter to be displaced in an inguinal hernia [57,58,59,60,61,62]. Statistically, females are much less likely to develop a herniation of the ureter in the inguinal region, where the most common appearance is the ureterocrural hernia [19,63,64].

Although trapping the ureter in the inguinal canal is a rare possibility [65], when confronted with an unexpected discovery, such as an ureteroinguinal hernia, extra caution should be maintained to ensure that no structures are accidentally identified harmed. Here, we attempt to determine the course of diagnosis for ureteroinguinal hernias, the most reliable and accessible diagnosis methods, as well as the case management by updating the existing literature with our experience. The current paper was reported in accordance with the SCARE criteria [66].

## 2. Discussion

### 2.1. Clinical Experience

A 75-year-old Caucasian male was brought to the emergency room complaining of gradually increasing high-intensity pain involving the right inguinal region and the scrotum, onsetting a week prior to admission. Anamnesis revealed a history of high blood pressure, chronic venous insufficiency, grade III obesity, chronic gastritis, and type II diabetes mellitus. The patient underwent multiple surgical interventions, including a trans-urethral resection of the prostate, varicocelectomy, umbilical hernia repair, cholecystectomy, and right total hip arthroplasty eight years prior to the respective hospital admission. A small size inguinal hernia was first noticed by the patient a short time after the hip replacement surgery. It was followed by multiple episodes of severe unilateral and colicky flank pain associated with urinary tract infections.

A review of systems showed a normal gastrointestinal tract with regular intestinal habits and no emetic or nauseous symptoms until hospital admission. The patient presented with abnormal genitourinary with dysuria, urinary frequency, urgency with weak urinary flow, and right flank pain, and no hematuria. The patient’s vitals were stable on clinical examination, and a large, irreducible inguinoscrotal hernia of about 20 cm located at the right scrotum was identified, having the superior pole at the deep inguinal ring descending into the right scrotum. The scrotum was swollen and painful spontaneously and on touch, having an increased consistency. Additionally, tight phimosis was observed. A complete blood test was within normal ranges, and urinalysis showed leukocyturia and microbial flora growing *E. coli*, which indicated antibiotic therapy. There was a slight distension of the pelvis and right ureter on ultrasound, without ultrasound-visible stones.

### 2.2. Diagnostic Features and Imaging Findings

An open surgery incision for repairing the inguinal hernia was performed under general anesthesia. On inspection, we identified a bulky fatty mass originating from the retroperitoneum. It protruded through a considerably large defect at the lateral inguinal fossa down to the right scrotum, tightly adherent to the external spermatic fascia. While dissecting the herniated adipose tissue, a white tortuous loop of about 10 cm long was discovered. Showing signs of peristalsis on stimulation, the first intention was to guide a 3F ureteral catheter on the right ureteral meatus. Due to the impossibility of completely reducing the herniated part of the ureter back into its anatomical position in the retroperitoneum, we decided to perform an 8 cm distal segmental ureterectomy with end-to-end anastomosis on the JJ tube (Figure 1, Figure 2 and Figure 3). On further exploration, there were no signs of a hernia sac; thus, suturing the abdominal wall was performed. Additionally, circumcision was performed, indicated by the tight phimosis. After surgery, the patient had favorable evolution, having the JJ tube removed six weeks past the intervention, allowing for physiological micturition.

### 2.3. Similar Findings

We identified eight case reports describing the diagnosis and surgical management of extraperitoneal ureteroinguinal hernias. In 2001, Giglio et al. [19] treated a 74-year-old patient with an asymptomatic inguinal hernia. The left ureteral herniation was an incidental finding since the patient underwent excretory urography for renal colic. The surgeons opted for a prolene prosthesis hernia repair after sparing the ureter and placing it back into the anatomical position. They described the ureter as normotrophic and normoperistaltic while being covered by adipose tissue without involving a peritoneal sac. In 2005, E Akpinar et al. [67] encountered a bilateral scrotal extraperitoneal herniation of the ureters in a 78-year-old patient from Turkey who developed pain and hematuria due to nephrolithiasis. The patient had bilateral scrotal enlargement during the examination, while the imagistic findings indicated bilateral ptotic kidneys with hydroureteronephrosis. A further CT urography confirmed the bilateral ureteral herniation, but the severe cardiac pathology contraindicated the surgical intervention. A 58-year-old patient [68] was admitted to the hospital in 2009 with the purpose of surgical intervention for a clinically diagnosed varicocele without swelling reduction while lying supine. Instead, during the intervention, the surgeons discovered the left ureter reaching the superior pole of the testis, forming a loop back into the peritoneal cavity, with no peritoneal sac adjacent to the ureter. After assessing the ureteral viability, the surgeons opted to reduce the hernia, although a second intervention due to symptomatic hydroureteronephrosis imposed the resection of the tortuous ureter with end-to-end anastomosis and uncomplicated recovery. Additionally, in 2009 E Golgor et al. [69] described the case of a 68-year-old patient with left-sided hydronephrosis, 18 years apart from a hernia repair on the same side. Contrast CT confirmed the ureteral herniation into the scrotum, while the case was managed with a 20 cm ureteral resection. Here, the extraperitoneal type was identified intraoperatively since there was no herniation sac involved. The next case of extraperitoneal ureteral hernia was identified in 2015 by Cetrulo et al. [70], where a 53-year-olde man previously had a left inguinal hernia repaired. He complained of difficulties urinating and a big protrusion in his left scrotum when he came to our clinic. As well as in the previous case reports, a CT urography revealed a massive hernia in the left scrotum, including the bladder and both ureters. The patient had laparoscopic repair with the assistance of a robotic system. The bladder and both ureters were removed carefully from the scrotum, while the testicle and spermatic cord were preserved. The defect was subsequently repaired using a pre-peritoneal polyester composite mesh. Similar to other cases, the patient required ureteral stents to allow for proper drainage, although, in this case, the use of a surgical robot allowed for an injury-free ureteral manipulation out from the scrotum.

In 2016, Di Nicolo et al. [71] reported that a 52-year-old man with a history of urinary tract infections and a prior clinical diagnosis of a left-sided inguinal hernia, presented to the nephrologist with the recent start of dysuria and the increased dimension of the left inguinoscrotal area in the absence of fever or scrotal injuries. Urogenital examination revealed a tiny palpable mass of the testis on the left, consistent with an inguinal hernia or hydrocele. A pre-operative ultrasound revealed an anechoic tubular shape with a dilated herniated ureter. Contrast-enhanced CT scans indicated that the more distal and proximal portions of the left ureter seemed normal, with an orthotopic opening in the bladder. The same ureter took an unusual path, passing via the inguinoscrotal area, where it grew in size to become a megaureter. Here, the radiologists were able to diagnose the extraperitoneal type clearly and suggested further surgical intervention to resolve the cause of recurrent urinary tract infections, but the outcome of this patient remains unknown. In 2017 a 78 years old male was admitted to the ICU with septic shock after complaining of urinary symptoms [72]. Considering he was observed as having a massive inguinal hernia on clinical examination, a CT scan uncovered a horseshoe kidney and the left ureter descending in the scrotum, with a stone obstructing the vesicoureteric junction. Considering the absence of abdominal contents descending into the scrotum and the anatomical variance of the kidneys, this case indicates an extraperitoneal ureteroinguinal hernia. Given the patient’s comorbidities, general surgery to correct the ureteroinguinal hernia was not performed, although an emergency stenting procedure to de-obstruct the urinary tract was needed. The most recent case [73] was brought in 2019, describing a 12year-old boy with a renal calculus and a history of cloacal exstrophy, cross-fused pelvic kidney, bladder neck reconstruction, and ileocystoplasty. He was further diagnosed upon clinical evaluation with an asymptomatic left-sided inguinal hernia without hydronephrosis on an ultrasound exam. A classic hernia repair procedure was performed to reveal the left ureter looping into the scrotum and no herniation sac. It was necessary to partially resect the left ureter and perform end-to-end anastomosis over a JJ stent, similar to our presented case.

Although most of the reported cases of ureteroinguinal hernia are diagnosed intraoperatively, under classic hernia repair and a high rate of ureteral damage due to unrecognizable diagnosis, the recent advances in imaging and surgical techniques allow for more precise diagnosis, preferably before surgical intervention for inguinal hernia, and facile management of these occurrences. Such was the case reported by Smith et al. [74] involving a 63-year-old male patient who underwent a CT scan to follow-up for partial nephrectomy after being diagnosed with renal cell carcinoma. Although his complaints were mild and correlated with the previous surgical intervention, the CT scan indicated hydroureteronephrosis, with a segment at the level of the inguinal canal. This scenario allowed for a pre-interventional diagnosis of the ureteral herniation through the inguinal canal, simplifying the surgical approach for a robotic-assisted laparoscopic ureterolysis, proving that urinary complaints associated with signs of inguinal hernia can predict the ureteral herniation, requiring ultrasound and CT imaging studies.

As was the case in our report, ureteral hernia might occur with or without obstructive symptoms, such as hydronephrosis. Even when hydronephrosis is present, the clinical picture may or may not show symptoms, such as ipsilateral flank discomfort. If the bladder is also implicated, the blockage of the bladder outlet may develop, producing symptoms of the lower urinary tract, such as frequency, urgency, and difficulty emptying. We believe the broader access and use of imaging techniques resulted in a greater number of ureteroinguinal hernias being diagnosed preoperatively. However, in the absence of evident bladder involvement or a urography phase highlighting the urinary system, the existence of ureteral herniation may be difficult to detect completely in the absence of hydronephrosis. Thus, when a hernia is discovered intraoperatively or when the repair is planned, surgeons should maintain a high index of suspicion that critical structures, including the ureter, may be present within the hernia or sac, especially in the setting of ipsilateral flank pain, renal failure, or hydronephrosis on the ultrasound.

On the contrary, a review published in 2021 argued the rarity of ureteral inguinal hernia and proposed a management algorithm [75]. The ureteric involvement in an inguinal hernia is indeed very rare, although recent progress in renal transplantation potentially modified the incidence of these hernias. As described in this review, due to the anterior placement of the kidney allograft in heterotopic kidney transplantation, ureteric incarceration into an inguinal hernia is anatomically plausible, which is proven by the increasing frequency of such reports. Two consecutive cases of men older than 60 years, with a history of an intermittent obstructive uropathy presenting with ureteroinguinal hernia after a kidney transplant, were described with a clinical profile demonstrating abdominal pain, gross hematuria, and acute kidney injury, two years and, respectively, 11 years after receiving the transplant. The abdominal ultrasound greatly contributed to the presumptive diagnosis. Decompression nephrostomy for hydronephrosis and nephrostogram revealed that the transplanted ureter was located below the external inguinal ligament, allowing for the CT scan to confirm the diagnosis of an ureteroinguinal hernia. Similar to our described images, the herniated ureters presented with a tortuous course and a U-turn towards the inguinal canal. Such findings should raise the concern of a ureteroinguinal hernia if accompanied by urinary symptoms.

It is critical to emphasize that most case reports depict a patient with symptoms of urinary obstruction [76,77,78,79,80], while it should be recognized that some individuals with transplant ureteric involvement inside an inguinal hernia are likely entirely asymptomatic. According to these data, an inguinal hernia with a transplanted ureter may be handled utilizing normal techniques. Once hydronephrosis is diagnosed through ultrasonography, the abdomen and groin should be examined for hernias; further imaging may include a nephrostogram, magnetic resonance imaging, or CT scan. After image-guided percutaneous nephrostomy tube installation to treat obstructive uropathy, an antegrade ureteric stent is implanted to allow for antegrade urine drainage and to help in the intraoperative localization and preservation of the transplanted ureter. In terms of the surgical strategy, an open ventral approach with an absorbable or biologic mesh is recommended, even more so when ureteric manipulation is undertaken. Distal ureteric resection and reimplantation should be reserved for patients who have had ureteric damage as a result of pre-existing stricture, pressure necrosis as a result of hernia incarceration, or incidental injury during surgery. The authors recommended retaining the pre-operative ureteric stent or placing one intraoperatively if the ureteric anastomosis is performed, followed by a postoperative nephrostogram prior to stent removal following surgery [75].

Compared to the existing literature, our case is remarkable as an inguinal hernia involving extraperitoneal ureteral elements in a patient with native kidneys located in their anatomical position, as the condition is prone to occur after renal transplant [81]. Most of the documented cases of extraperitoneal inguinal hernias were incidentally diagnosed, where the most common finding was hydronephrosis on the affected side. A CT urography was performed in many of the described cases, which allowed for a certified diagnosis. The patients suffered ureteral damage to a certain degree, suggesting urological management after the hernia was repaired, or the ureteral elongation required resection to prevent a recurrence. The diagnosis of ureteral hernia in an inguinoscrotal hernia is usually overlooked due to most patients’ lack of urinary symptoms. A thorough urological history should cover any inguinal hernia to certify a potential bladder or ureteric involvement. Additional imaging should be performed when an inguinal hernia is suspected based on the patient’s history and physical exam. Ultrasonography alone was proven inefficient in diagnosis [82]; thus, a CT scan or intravenous pyelogram can be performed to validate the diagnosis. Surgery should be considered to prevent bladder, ureteric, and renal complications, and to resolve the presenting symptoms.

## 3. Conclusions

Summarizing the data available in the literature, the ureteroinguinal hernia is a rare occurrence. Recently, this pathology has been diagnosed more frequently, secondary to kidney transplantation as an obscure cause of hydronephrosis. In such cases, abdominal repositioning is advisable. Primary ureteroinguinal hernia is so uncommon that less than 10 cases have been reported in the adult population. In the presence of urinary symptoms, urographic CT usually confirms the diagnosis. On the other hand, although rare in incidence, the possibility must be considered while operating on a massive sliding hernia on the right side. If the ureter is identified under such circumstances, the resection followed by the aureteral reconstruction on a JJ stent is considered as a safe decision.

## Figures and Tables

**Figure 1 diagnostics-12-00353-f001:**
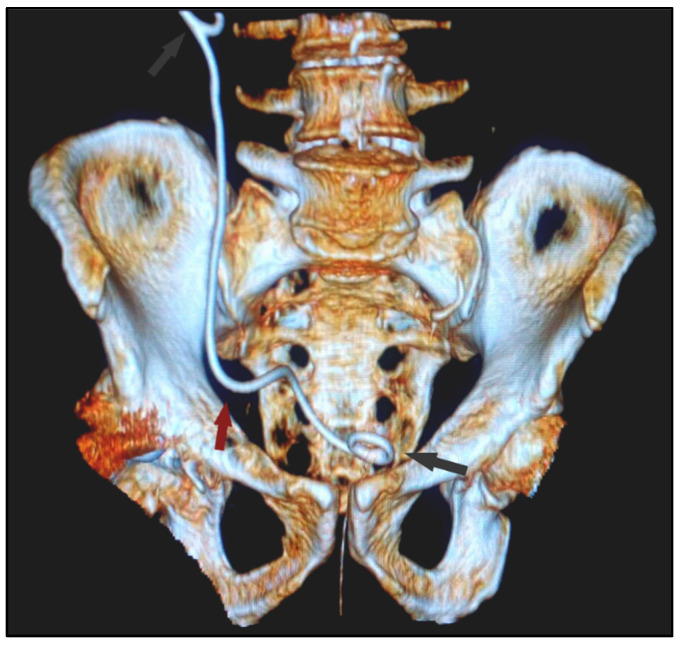
3D reconstruction of the postoperative pelvic CT scan with urography in a 75-year-old male with native kidneys and a right JJ Stent placement (curled ends highlighted by gray arrows). Bulging of the inferior 1/3 of the ureter (red arrow) towards the inguinal canal.

**Figure 2 diagnostics-12-00353-f002:**
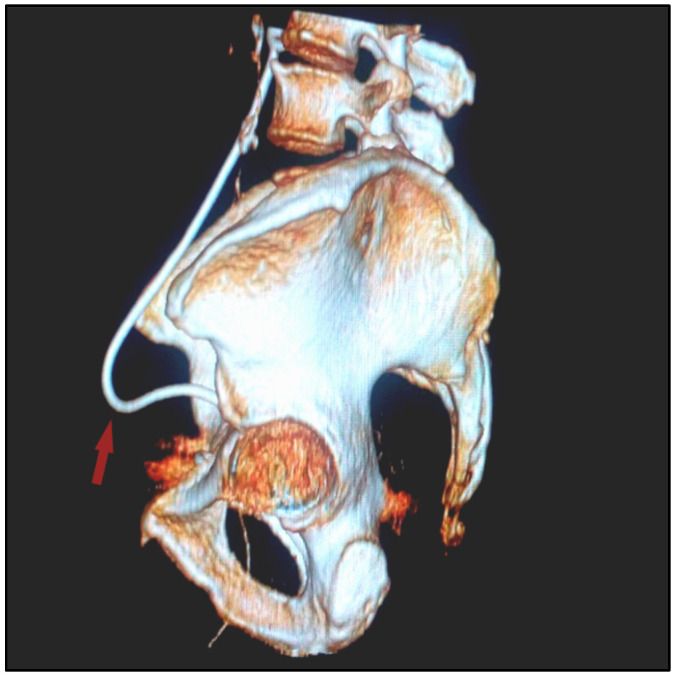
3D reconstruction of the postoperative pelvic CT scan with urography. Left-lateral view of the bulged inferior 1/3 of the ureter (red arrow) towards the inguinal canal.

**Figure 3 diagnostics-12-00353-f003:**
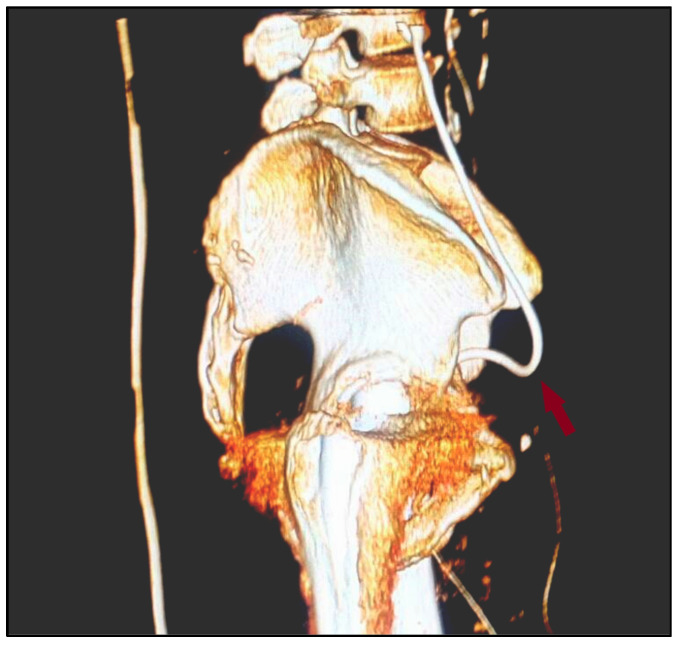
3D reconstruction of the postoperative pelvic CT scan with urography. Right-lateral view of the bulged inferior 1/3 of the ureter (red arrow) towards the inguinal canal.

## Data Availability

The data presented in this study are available on request from the corresponding author.

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
