# Peer review of "Difficulties in Diagnosing Extraperitoneal Ureteroinguinal Hernias: A Review of the Literature and Clinical Experience of a Rare Encounter in Acute Surgical Care Settings"

_diagnostics, 2022, doi:10.3390/diagnostics12020353_

Round 1

Reviewer 1 Report

This is an interesting case report and review of the global literature on a very rare condition (ureteroinguinal hernia).

It should be published. However, the number of authors for a case report on a very rare condition and review of the literature is too high (nine (!) authors). The number should be reduced to maximally five authors. It is hard to believe that there was a substantial contribution of each author. 

Author Response

Dear reviewer,

Thank you for contributing with valuable feedback meaning to help us improve the quality of our research. After careful consideration of the comments received, we have made the following changes to our manuscript:

Regarding your concern towards the number of authors listed, in this case nine, we have to mention that although not all of them had an equal contribution to writing this manuscript, they all contributed significantly to the management of the patient described in our report. There was a mixed surgical and urology team who attended the actual intervention, as well as for the imaging studies. They all showed great interest into making this case available to the world of science, and we believe it is honest to mention all nine authors for their contribution, since the publisher doesn’t have a maximum requirement for authors on a review paper.

Best regards,

The authors

Reviewer 2 Report

Generally, the manuscript is all too long with too much text on generalities which h may be relevant in a textbook but not in research. Several paragraphs throughout all the manuscript may be omitted. The authors should focus upon the 6 reported case of surgical treatment for the condition and with a comparison to their own case. The cases should not be referred in detail. Why did the authors resect the ureter and not just repositioning it to the abdomen? Some of the references are wrongly cited. The figures are good and illustrative.   

Author Response

Dear reviewer,

Thank you for contributing with valuable feedback meaning to help us improve the quality of our research. After careful consideration of the comments received, we have made the following changes to our manuscript:

  • Regarding your concern towards the lengthy manuscript, we were previously suggested by the assistant editor to increase the size of the literature review, to comprise a wider description of studies reporting the extraperitoneal ureteroinguional hernia, and challenges faced in diagnosing and management of those occurrences, since the data is very limited by their rarity.
  • Regarding the ureteral resection, we already mentioned at lines 152-154 that “Due to the impossibility of completely reducing the herniated part of the ureter back into its anatomical position in the retroperitoneum we decided to perform an 8 cm distal segmental ureterectomy”
  • As you mentioned “several paragraphs throughout all the manuscript may be omitted” – please describe what are those paragraphs and the reason for omitting them. Also, “Some of the references are wrongly cited” – we checked again the citations but couldn’t see wrong citations. Please name them to be able to revise.

Best regards,

The authors

Reviewer 3 Report

The paper describes an interesting case of ureter herniation in a case of inguinal hernia that it is so uncommon that deserve to be reported in literature.

The paper is well written and has wonderful 3D CT scan reconstruction.

By the way, as a surgeon, I think that intra operative pictures are mandatory especially if authors describe extremely rare findings. Therefore I would ask you to add some pictures from surgery in order to see the ureter herniation if available.

Author Response

Dear reviewer,

Thank you for contributing with valuable feedback meaning to help us improve the quality of our research, as we appreciate your opinion.

Unfortunately, we don’t have any pictures from surgery to complement the 3D CT reconstructions.

There were no other specific changes made during this round of review, besides several formatting changes in the references section.

Best regards,

The authors

Round 2

Reviewer 2 Report

As I understand it, most cases of ureteral herniation are in patients with renal transplantation where the etiology, treatment and prognosis must be different from the incidental finding of an ureteral herniation in patients without urological pathology, and the manuscript should focus upon these cases, only.

When it comes to references, it is stated in line 61 that less than 150 cases have been reported in the literature with a reference to a genetical study in pigs (ref 4)?

In line 62 it is stated that “around” 10 cases of ureteral herniation have been published. In line 170 it is stated that 8 cases with a surgical treatment have been published. The 8 cases are described in detail and 2 of them were not operated! A summarized description of the cases could be more appropriate to my opinion.

The CT scans does not show the herniation of the ureter but only a “bulging towards the inguinal canal”.  The authors should comment on this.

The conclusion is far too long and more precise recommendation on treatment. Must we aim for a reposition to the abdomen or an ureteral resection?

Author Response

Dear reviewer,

Thank you again for your contribution towards improving our manuscript. Please consider the following comments based on your feedback.

  1. As shown in the article's Discussion section, there are two main particular types of inguinoureteral hernia. The one which is less frequent, produced by primary mechanism, such as embryological development disorders, and the other, a little bit more often encountered, secondary to kidney transplantation. Being a literature review based upon a case presentation of a primary inguinoureteral hernia, we considered useful to discuss about both mechanisms as an awareness for practitioners regarding the possibility of such an occurrence encountering. In our opinion, both possibilities are likely to occur, so both should be discussed.
  2. There has been a typing mistake corroborating the text with the cite. In order to correct it, we modified both the text and the citation ( less than 140 cases, according to reference number 17 that we changed in the references section).
  3. Citation the bibliographic source number 18 ( Won Ac et al.) there are roughly around 10 cases of inguinoureteral hernia due to primary causes cited in the literature. On the other hand, we succeeded in identifying only 8 of them, with wide descriptions of clinical and imagistic assessment and operative management. To avoid further misunderstanding, we should mention only the 8 cases we were able to document. Considering the very small number of cases, heterogenous presentation, and particular therapeutic solutions, we considered it useful to present each case briefly as its authors depicted it.
  4. As described in the case presents itself, in our circumstances, the inguinoureteral hernia was discovered intraoperatively, incidentally. We did not perform any preoperative imagistic explorations. The CT scan reconstructions belong to the postoperative assessment, after the segmental uretherectomy with end-to-end anastomosis was carried out. Due to this, only the bulging of the ureter is visible on imagistic. Pictures description and case report itself were readjusted, in order to emphasize this, in a much clear manner.
  5. In response to your suggestion, the conclusion was re-edited in a more concise fashion, emphasizing different types of mechanism and clinical scenario, providing diagnosis and treatment recommendations in accordance.

Best regards

Round 3

Reviewer 2 Report

Thank you for the sufficient reply and the revisions, that has improved the manuscript considerably.